# TRIPLET SIMILARITY LEARNING ON CONCORDANCE CONSTRAINT

## ABSTRACT

Triplet-based loss functions have been the paradigm of choice for robust deep metric learning (DML). However, conventional triplet-based losses require carefully tuning a decision boundary, *i.e.*, violation margin. When performing online triplet mining on each mini-batch, choosing a good global and constant prior value for violation margin is challenging and irrational. To circumvent this issue, we propose a novel yet efficient concordance-induced triplet (CIT) loss as an objective function to train DML models. We formulate the similarity of triplet samples as a concordance constraint problem, then directly optimize concordance during DML model learning. Triplet concordance refers to the predicted ordering of intra-class and inter-class similarities being correct, which is invariant to any monotone transformation of the decision boundary of triplet samples. Hence, our CIT loss is free from the plague of adopting the violation margin as a prior constraint. In addition, due to the high training complexity of triplet-based losses, we introduce a partial likelihood term for CIT loss to impose additional penalties on hard triplet samples, thus enforcing fast convergence. We extensively experiment on a variety of DML tasks to demonstrate the elegance and simplicity of our CIT loss against its counterparts. In particular, on face recognition, person re-identification, as well as image retrieval datasets, our method can achieve comparable performances with state-of-the-arts without tuning any hyper-parameters laboriously.

## 1 INTRODUCTION

Deep metric learning (DML) for visual understanding tasks, *e.g.*, face recognition Schroff et al. (2015); Taigman et al. (2014), person re-identification (ReID) Shi et al. (2016); Ustinova & Lempitsky (2016), image retrieval Fang et al. (2021); Revaud et al. (2019), aims at learning embedding representations of images with class-level labels by a ranking loss function Kaya & Bilge (2019); Sohn (2016); Wang et al. (2017). There are two representative ranking loss functions developed for DML to minimize between-class similarity and maximize within-class similarity, *i.e.*, pair-based loss Sun et al. (2014) and triplet-based loss Zhao et al. (2019). Compared to pairwise constraints, the optimization pattern of triplet-based losses additionally captures the relative similarity information, thus yielding impressive performances Liang et al. (2021); Zhuang et al. (2016). With triplet constraints, images from the same class are projected into neighboring embedding spaces, and images with different semantic contexts are mapped apart. However, under such an optimization objective, triplet-based losses suffer from following two problems when training DML models with the stochastic gradient descent (SGD) algorithm and sampling triplets within a mini-batch.

•**Irrational to set an absolute margin.** Triplet constraint relies on a decision boundary to partition the embedding space of intra-class and inter-class, *i.e.*, violation margin for reinforcing optimization Wang et al. (2018a;b). However, the violation margin is sensitive to scale change, and choosing an identical absolute value for clusters in different scales of intra-class variation is inappropriate Wang et al. (2017). Hence, triplet-based losses need to regulate this hyper-parameter attentively to impose appropriate penalty strength Qian et al. (2019); Sun et al. (2020). The performance of Circle loss Sun et al. (2020) on the varying circular decision boundary can prove such a claim. The performance of the same task exhibits a significant difference by setting different violation margins. And Circle loss with the same violation margin varies from superior to inferior on various tasks. For circumventing this issue, Angular loss is proposed to push the negative point away from the center of the positive cluster and drag the positive points closer to each other by constraining

the upper bound of the angle at the negative point Wang et al. (2017). In hierarchical triplet loss (HTL) Ge (2018), the violation margin is automatically updated over the constructed hierarchical tree to identify a margin that generates gradients for violated triplets. However, existing methods of mitigating the issue still depend on the setting of the decision boundary, only substituting a hyperparameter. Angular loss needs to specify the angle degree and HTL needs to design a hierarchical class tree. Since choosing a global and constant prior value for the decision boundary is irrational, we innovatively formulate triplet similarity learning as a concordance constraint problem without an assumed decision boundary.

•**Suffering from slow convergence.** Triplet-based losses can provide a strong supervisory signal for training DML models by mining rich and fine-grained inter-sample relations. However, since the number of tuples (each tuple contains an anchor sample and its positive and negative samples) increases polynomially with the number of training samples, they suffer from prohibitively high training complexity, thus causing significantly slow convergence Ebrahimpour et al. (2022); Kim et al. (2020). Another potential issue for triplet-based losses is that a large amount of tuples make a limited contribution to the learning algorithm and sometimes even diminishes the quality of the learned embedding space Wu et al. (2017). Many works have been devoted to studying the effective triplet sampling strategy within a mini-batch to utilize hard triplet samples that improve convergence speed or the final discriminative performance Hermans et al. (2017); Oh Song et al. (2016); Sohn (2016); Wu et al. (2017). For example, HTL Ge (2018) is proposed to automatically collect informative training triplets via an adaptively-learned hierarchical class structure. However, these hard triplet sample mining techniques involve tuning hyper-parameters and may occur the risk of overfitting when performing online triplet mining within a mini-batch Ebrahimpour et al. (2022); Kim et al. (2020). Given three tuple types (hard, semi-hard, and easy triplet samples), we need to consider how to achieve the trade-off between them during DML optimization. Leveraging hard triplet samples alone may occur bad local minima Do et al. (2019). Overwhelming easy triplet samples affect the training efficiency Schroff et al. (2015). Inspired by SoftTriplet loss Qian et al. (2019) introduced to learn embeddings without triplet sampling, we explore laying more emphasis on hard triplet samples by relaxing concordance constraints, thus accelerating convergence speed.

In each triplet sample, the intra-class similarity is naturally higher than the one of inter-class. The predicted ordering of similarities of intra-class and inter-class needs to be on par with the observed ordering. Such an ordering concordance not only takes effect on a mini-batch but also on the whole sample. Such intrinsic concordance constraint is invariant to any monotone transformation of the decision boundary of triplet samples. Hence, we develop a novel concordance-induced triplet (CIT) loss function to optimize triplet similarity. Existing triplet-based losses explicitly give a global and constant violation margin as a decision boundary based on apriori knowledge. Unlike them, our CIT loss exploits the concordance constraint of triplet similarity to avoid falling into the plague of tuning the violation margin. It is an elegant, simple, and efficient way to learn the intrinsic similarity between all samples and is insensitive to the triplet sampling within a mini-batch. We further introduce a partial likelihood term to enforce different penalty strengths on different tuple types, primarily laying more penalties on hard triplet samples. This term mainly helps improve convergence speed and exhibits a slight impact on performance, thus avoiding the plague of elaborative tuning. Based on thoroughly and randomly mini-batches and triplet sampling, this term can regulate the penalty strength keeping consistency with the degree of the discordance or concordance of triplet similarity. The higher discordance of hard triplet samples brings more penalty strengths, thus arising more contributions to gradients.

The main contributions of this work are summarized as follows:

- We propose a novel, simple, elegant concordance-induced triplet (CIT) loss function for deep metric learning (DML). Our CIT loss frees DML training from tuning the decision boundary by directly maximizing concordance of triplet similarity.

- In addition, we introduce a partial likelihood term to impose loose concordance constraints to focus on the informativity of hard triplet samples, thus helping speed up convergence.

- Using two popular backbones, we conduct extensive experiments on various DML tasks, including face recognition, person re-identification (Reid), and image retrieval. On all tasks, we demonstrate the effectiveness and elegance of our CIT loss and gained performance on par with state-of-the-art.

## 2 METHODOLOGY

### 2.1 WARM-UP

Given a training set $\mathcal{D} = \{(\boldsymbol{x}_i, y_i)\}_{i=1}^{N}$ with $K$ classes, $\boldsymbol{x}_i$ with label $y_i \in \{0, 1, \cdots, K-1\}$ denotes the feature embedding projected from the $i$-th image with DML models. Let the feature embedding of $\boldsymbol{x}_i \in \mathbb{R}^D$ be represented as $\mathcal{F}_{\boldsymbol{\theta}}(\boldsymbol{I}_i)$, where $\boldsymbol{I}_i$ is the $i$-th input image, $\boldsymbol{\theta}$ indicates the learnable parameters of a differentiable DML model $\mathcal{F}$, and $D$ is the dimension of feature embedding. With DML model $\mathcal{F}$, we can map the image space $\boldsymbol{I}_i$ into low-dimension embedding space $\boldsymbol{x}_i$ used for similarity measurement. $\boldsymbol{x}_i$ is usually normalized into unit length for the training stability and comparison simplicity. DML models primarily utilize various ranking loss functions to learn the embedding space from the image space in a supervised way. Apart from a suitable neural network, it is essential to design a ranking loss function to optimize the embedding space. Our work aims to improve triplet-based loss to help DML models train elegantly and efficiently by avoiding tuning the violation margin and improving convergence speed, and can gain impressive performance.

DML models are commonly trained using online SGD algorithms, where the gradients for optimizing network parameters are computed locally with mini-batches. Hence, triplet samples are selected and formed in a mini-batch during each training iteration. Each triplet sample $\mathcal{T} = (\boldsymbol{x}_a, \boldsymbol{x}_p, \boldsymbol{x}_n)$ consists of an anchor sample $\boldsymbol{x}_a$, a positive sample $\boldsymbol{x}_p$ and a negative sample $\boldsymbol{x}_n$, whose labels satisfy $y_a = y_p \neq y_n$. The goal of triplet-based losses is to push away the negative sample $\boldsymbol{x}_n$ from the anchor sample $\boldsymbol{x}_a$ by a violation margin $m > 0$ compared to the positive sample $\boldsymbol{x}_p$:

$$S_{an} + m \leq S_{ap}, \tag{1}$$

We define $S_{ap} \in [0, 1]$ as the intra-class similarity of $\boldsymbol{x}_a$ and $\boldsymbol{x}_p$, and $S_{an} \in [0, 1]$ as the inter-class similarity of $\boldsymbol{x}_a$ and $\boldsymbol{x}_n$. We seek to minimize $S_{an}$ and maximize $S_{ap}$. To enforce this constraint in the embedding space, we define the optimization target of the standard triplet loss as:

$$\mathcal{L}_{st} = \frac{1}{N_{\mathcal{T}}} \sum_{\mathcal{T}} \left[ S_{an} - S_{ap} + m \right]_{+}, \tag{2}$$

where the operator $[\cdot]_+ = max(0, \cdot)$ represents the hinge function and the symbol $N_{\mathcal{T}}$ denotes the number of all triplet samples in a mini-batch.

According to Equation 2, given a globally constant value for the violation margin $m$, we can group all triplet samples into three categories:

- **Hard triplet samples:** if $S_{an} > S_{ap}$,
- **Semi-hard triplet samples:** if $S_{ap} - m < S_{an} < S_{ap}$,
- **Easy triplet samples:** if $S_{an} + m < S_{ap}$.

Among these three tuple types, easy triplet samples generate zero loss, while hard triplet samples contribute the most losses. An effective sampling strategy combining an appropriate violation margin can help mine hard triplet samples. In fact, the violation margin in the triplet-based losses plays a key role to sample selection during model training Ge (2018). However, the violation margin needs to be carefully tuned. On the other hand, the absolute violation margin is an irrational decision boundary for multi-classes with different cluster centroids.

### 2.2 CIT LOSS

Being simple, we intuitively explore no need to consider the violation margin for learning hard triplet samples effectively. Since the concordance is invariant to any monotone transformation of the decision boundary of triplet samples, it is natural to formulate the metric learning problem to maximize the concordance. We turn the predicted similarity correlation of each triplet sample $\mathcal{T}$ into a comparable pairs, *i.e.*, intra-class similarity $S_{ap}$ and inter-class similarity $S_{an}$. The set of comparable pairs mined from the whole training set $\mathcal{D}$ is $\mathrm{E}_{\mathcal{T}} := \{(S_{ap}, S_{an})\}$. The number in this set is $N_{\mathcal{T}}$. A comparable pair $(S_{ap}, S_{an}) \in \mathrm{E}_{\mathcal{T}}$ is concordant if $S_{ap} > S_{an}$. Otherwise, the pair is discordant with the ground truth. We calculate the ratio of undergoing any pairs by taking the exponential form of intra-class and inter-class similarities:

$$R = \frac{1}{N_{\mathcal{T}}} \sum_{(S_{ap}, S_{an}) \in \mathrm{E}_{\mathcal{T}}} \frac{e^{S_{an}}}{e^{S_{ap}}}. \tag{3}$$

Obviously, the average ratio $R$ lies in the range $[e^{-1}, e]$. The lower and upper ranges correspond to complete concordance and discordance of all comparable pairs, respectively. With such bounds, we define our CIT loss as an exponential lower-bound form:

$$\mathcal{L}_e = \frac{1}{N_\mathcal{T}} \sum_{(S_{ap}, S_{an}) \in \mathrm{E}_\mathcal{T}} \left[ 1 - e^{-(S_{an} - S_{ap})} \right]_+. \tag{4}$$

With minimizing CIT loss, we address the metric learning problem as concordance optimization between the predicted and observed similarity of comparable pairs. Similarity concordance building in whole triplet samples is invariant to any monotone transformation of the decision boundary of triplet samples. In other words, concordance optimization pays emphasis on the ordering correlation of distance without considering the correlation degree, thus avoiding imposing margin constraints.

The empirical error induced by pairwise discordance with respect to DML model $\mathcal{F}_{\boldsymbol{\theta}}$ is denoted by $\mathcal{G}(\mathcal{F}_{\boldsymbol{\theta}})$ and defined by:

$$\mathcal{G}(\mathcal{F}_{\boldsymbol{\theta}}) = \frac{1}{N_\mathcal{T}} \sum_{(S_{ap}, S_{an}) \in \mathrm{E}_\mathcal{T}} \mathbb{I}_{S_{ap} < S_{an}} \geq \mathcal{L}_e, \tag{5}$$

where the indicator function $\mathbb{I} = 1$ if $S_{ap} < S_{an}$ (discordance), and 0 otherwise (concordance). According to Equation 5, we can estimate the learning parameters $\theta$ of the DML model $\mathcal{F}$ by minimizing CIT loss $\mathcal{L}_e$. We leverage concordance-induced penalty in our CIT loss to optimize the target pairwise similarity.

We can assume that the violation margin $m$ is 0 in our CIT loss, then there are two tuple types: hard triplet samples if $S_{ap} < S_{an}$ and easy triplet samples if $S_{ap} > S_{an}$. For easy triplet samples, the DML model $\mathcal{F}$ faultlessly predicts the target pairwise ranking, thus leaving penalty-free. And hard triplet samples contribute to the informativity for gradient-based optimization. To speed up convergence, we further introduce a partial likelihood term for our CIT loss to focus on the discordant penalty of hard triplet samples. From the standpoint of concordance, the pairwise similarity in each triplet sample $\mathcal{T}$ is not always transitive. The transitivity of concordance-induced order by the DML model $\mathcal{F}_{\boldsymbol{\theta}}$: $S_{ap} > S_{an}$ but may $S_{ap} < S_{pn}$ for each triplet sample $\mathcal{T}$. $S_{pn}$ indicates inter-class similarity between $\boldsymbol{x}_p$ and $\boldsymbol{x}_n$. Considering transitivity of triangle edge, we define the partial likelihood form as:

$$\mathcal{L}_p = \prod_{(S_{ap}, S_{an}) \in \mathrm{E}_\mathcal{T}} \frac{e^{S_{ap}}}{e^{S_{an}} + e^{S_{pn}}}. \tag{6}$$

Theoretically, the higher proportion of the edge $S_{ap}$ in the three sides can better account for the concordance and transitivity. Hence, the product of the ratio in $\mathcal{L}_p$ can suggest the degree of concordance or discordance. We quantize such a degree to construct a loose concordance constraint term by placing the negative log partial likelihood of Equation 6:

$$\mathcal{L}_p = -\frac{1}{N_\mathcal{T}} \sum_{(S_{ap}, S_{an}) \in \mathrm{E}_\mathcal{T}} \left\{ S_{ap} - \log(e^{S_{an}} + e^{S_{pn}}) \right\}. \tag{7}$$

With Equation 7, the penalty on predictive error for hard triplet samples can be boosted, thus helping gradient optimization. And the best predictive outcome for easy triplet samples is $S_{ap} \geq \log(e^{S_{an}} + e^{S_{pn}})$, there is no penalty. If not, the slight penalty for easy triplet samples can help enhance the clustering effect within the class. It can be seen from this that this term is subject to the principles of maximizing inter-class similarity and minimizing intra-class similarity.

Combining Equations 4 and 7, we present our CIT loss as:

$$\mathcal{L}_{cit} = \gamma \mathcal{L}_e + (1 - \gamma) \mathcal{L}_p, \tag{8}$$

where the hyper-parameter $\gamma \in [0, 1]$ regulates the loss value between them. Specifically, $\gamma$ controls the magnitudes of the two losses at the same level to stabilize the model training. By comparing the two terms in our CIT loss, we can find that the former term requires a more rigorous concordance. However, it is impossible to ensure the complete transitivity and concordance. Hence, we can not fully replace the former term with the latter term. Generally, $\gamma = 0.5$ can trade off the discordant penalty and fast convergence. Schematically, (a) and (b) in Figure 1 show that our CIT loss can bring fast convergence by utilizing the hyper-parameter $\gamma$ to boost the contribution of hard triplet samples to gradient update of network parameters. And it helps reduce the training consumption of easy triplet samples, thus reaching up to the last performance in advance.

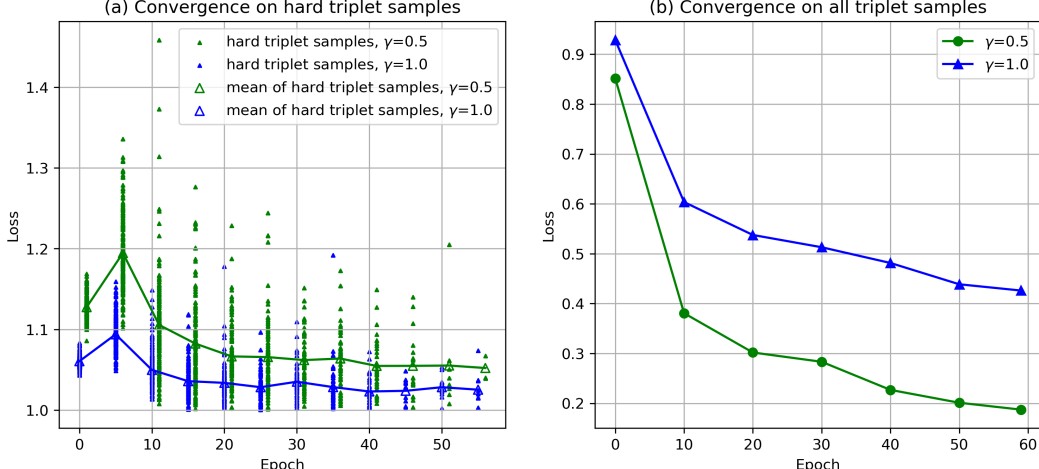

Figure 1: (a) For hard triplet samples, as the epoch increases, their number significantly decreases, and the average loss on $\gamma = 0.5$ (heavier penalties) gradually approaches $\gamma = 1.0$. (b) The convergence speed on $\gamma = 0.5$ is faster than $\gamma = 1.0$ due to imposing more penalties on hard triple samples.

## 3 EXPERIMENTS

### 3.1 SETTINGS

**Datasets.** (1) Person ReID aims to spot the appearance of the same person in different cameras or the same camera on different occasions. We evaluate our method on a popular dataset, Market1501 Zheng et al. (2015), containing 1,501 identities, 12,396 training images, and 19,732 gallery images captured with six cameras. (2) We use two datasets for evaluation on image retrieval, *i.e.*, Car196 Krause et al. (2013) and In-shop Clothes Liu et al. (2016). The Cars196 dataset is composed of 16,185 car images of 196 classes. And the In-shop Clothes Retrieval dataset has 11,735 classes of clothing items and 54,642 images. (3) The CASIA-WebFace dataset Yi et al. (2014) with 10,757 real identities and 494,441 face images is the most popular dataset for the training of face recognition. For evaluation, we adopt the face verification results on LFW Huang et al. (2008), AgeDB Moschoglou et al. (2017), IJB-C Maze et al. (2018), and CFP-FP Sengupta et al. (2016) datasets.

**Comparable methods.** Five comparable triplet-based loss functions involve (1) standard triplet loss (Triplet) with a violation margin representing the difference between the anchor-positive distance and the anchor-negative distance Schroff et al. (2015), (2) Angular loss with an angular prior used for separating different classes Wang et al. (2017), (3) centroid triplet (CT) loss with a margin hyper-parameter used as the decision boundary of centroids of positive and negative classes Wieczorek et al. (2021), (4) Circle loss with a relaxation margin used for controlling the radius of the decision boundary Sun et al. (2020), and (5) soft triple (ST) loss without triplet sampling and violation margin Qian et al. (2019). Both ST and our CIT are exempt from the violation margin setting. Our CIT leverages the intrinsic concordance between the predictive and observed similarities of triplet samples, while ST extends softmax loss with multiple centers for each class.

**Implement details.** We implement all loss functions on the pytorch-metric-learning Musgrave et al. (2020) platform and experiment with them on two different network structures. Two networks are convolutional neural network (CNN) of ResNet50 He et al. (2016) and vision transformer (ViT) Dosovitskiy et al. (2020). We set the hyper-parameters for comparable methods according to the default reported in their works, and the default of the hyper-parameter $\gamma$ in our CIT loss is 1.0, if not specified. We extract the 512-D feature embeddings for computing distances and use Euclidean distance as the metric during inferences. We adopt FastReID He et al. (2020) platform to train the two networks with different loss functions for three DML tasks. The CNN optimizer is Adam Kingma & Ba (2014), with a learning rate of 3.5e-4, while ViT is SGD, with a learning rate of 8e-3.

Table 1: Comparison of different loss functions for person ReID task on CNN and ViT networks in terms of rank-k (k=1,5,10, in %) accuracy, mean average precision (mAP, in %), and mean inverse negative penalty (mINP, in %).

| Method | Margin | Network | Market1501 | | | | |
|---|---|---|---|---|---|---|---|
| | | | $R@1$ | $R@5$ | $R@10$ | $mAP$ | $mINP$ |
| **Triplet** | $m = 0.01$ | CNN | 95.58 | 98.28 | 98.87 | 89.08 | 67.05 |
| | $m = 0.05$ | | 95.75 | 98.52 | 98.87 | 89.26 | 67.48 |
| | $m = 0.1$ | | 95.75 | 98.49 | 98.99 | 89.19 | 66.65 |
| **Angular** | $\alpha = 20$ | | 95.61 | 98.37 | 99.08 | **89.74** | **69.56** |
| | $\alpha = 40$ | | 94.98 | 98.13 | 99.05 | 87.29 | 62.75 |
| | $\alpha = 60$ | | 67.87 | 84.02 | 89.54 | 61.87 | 36.15 |
| **CT** | $m = 0.01$ | | 95.52 | 98.37 | 98.90 | 88.68 | 66.11 |
| | $m = 0.05$ | | 95.61 | 98.31 | 99.02 | 89.05 | 66.28 |
| | $m = 0.1$ | | 95.16 | 98.22 | 98.96 | 88.65 | 66.27 |
| **Circle** | $m = 0.1$ | | 92.10 | 96.82 | 97.60 | 81.84 | 52.83 |
| | $m = 0.4$ | | 91.18 | 96.59 | 97.71 | 79.65 | 48.91 |
| | $m = 0.6$ | | 93.05 | 97.54 | 98.40 | 82.37 | 52.97 |
| **ST** | - | | 95.43 | 98.34 | 98.87 | 88.66 | 66.04 |
| **CIT (Ours)** | - | | **95.87** | 98.22 | 98.93 | 89.41 | 68.27 |
| **Triplet** | $m = 0.01$ | ViT | 93.62 | 98.10 | 99.08 | 85.07 | 58.88 |
| | $m = 0.05$ | | **94.06** | 98.40 | 99.11 | 85.87 | 60.74 |
| | $m = 0.1$ | | 93.68 | 98.01 | 99.17 | 86.02 | **61.94** |
| **Angular** | $\alpha = 20$ | | 93.47 | 97.71 | 98.57 | 85.35 | 60.70 |
| | $\alpha = 40$ | | 91.95 | 96.47 | 97.86 | 80.68 | 52.74 |
| | $\alpha = 60$ | | 90.38 | 96.26 | 97.65 | 78.95 | 51.01 |
| **CT** | $m = 0.01$ | | 92.19 | 97.71 | 98.96 | 81.75 | 51.91 |
| | $m = 0.05$ | | 92.58 | 97.60 | 98.96 | 82.27 | 52.87 |
| | $m = 0.1$ | | 93.38 | 97.98 | 98.93 | 83.00 | 54.31 |
| **Circle** | $m = 0.1$ | | 87.68 | 93.88 | 95.46 | 72.80 | 42.64 |
| | $m = 0.4$ | | 93.32 | 97.71 | 98.46 | 85.13 | 61.62 |
| | $m = 0.6$ | | 92.37 | 97.92 | 98.93 | 80.90 | 49.63 |
| **ST** | - | | 91.81 | 97.51 | 98.78 | 80.69 | 50.00 |
| **CIT (Ours)** | - | | 93.88 | 97.74 | 98.93 | **86.23** | 61.88 |

Specifically, the learning rate is decay scheduled according to cosine annealing strategy Loshchilov & Hutter (2016). The input size for CNN is $384 \times 128$ and ViT is $256 \times 128$. The batch size of both CNN and ViT is $64$. The parameters of CNN are optimized in $60$ epochs, while ViT is $120$ epochs.

## 3.2 PERSON RE-IDENTIFICATION

We evaluate comparable loss functions on the ReID task in Table 1. We can make three observations from the reported performances. First, we can find that our CIT can achieve competitive performances against the state-of-the-art. CIT obtains the best (95.81) on CNN and the second-highest (93.88) on ViT in terms of $R@1$. And CIT achieves the best (86.23) on ViT and the second-highest (89.41) on CNN in terms of $mAP$. Moreover, $mINP$ of our CIT are on par with the second-highest methods (underline), showing the competence of staying in the first tier among all methods. Second, we report the performances on varying violation margins for those loss functions relying on the decision boundary (See column Margin in Table 1). We can discern that different violation margins can bring significant differences in performance, and the most prominent are Angular and Circle. Angular achieves the best $mAP$ and $mINP$ on CNN when the angle specified in degrees ($\alpha$) is 20 (its default is 40). But when $\alpha = 60$, the performance of Angular is pretty bad. In particular, we conduct repeated experiments to verify the reliability of these results. Such results entail that it is challenging to choose an appropriate violation margin. Third, both ST and our CIT are free from the plague of setting the violation margin. In terms of three key metrics, $R@1$, $mAP$, and $mINP$ on CNN and ViT, our CIT exhibits significant advantages against ST. In fact, there is a crucial hyper-

Table 2: Comparison of different loss functions for image retrieval tasks on CNN and ViT networks in terms of rank-k (k=1,5,10, in %) accuracy and mAP (in %).

| Method | Network | Cars196 | | | | In-shop Clothes | | | |
|---|---|---|---|---|---|---|---|---|---|
| | | $R@1$ | $R@5$ | $R@10$ | $mAP$ | $R@1$ | $R@5$ | $R@10$ | $mAP$ |
| **Triplet** | | 90.85 | 96.84 | 98.32 | **51.08** | 93.37 | 97.80 | 98.52 | 78.47 |
| **Angular** | | 89.36 | 96.70 | 97.97 | 43.12 | 90.34 | 95.91 | 97.00 | 71.71 |
| **CT** | **CNN** | 89.40 | 96.90 | 98.03 | 42.56 | **93.75** | 97.87 | 98.56 | **79.17** |
| **Circle** | | 87.05 | 95.93 | 97.27 | 41.26 | 87.81 | 94.90 | 96.51 | 70.00 |
| **ST** | | 90.05 | 97.04 | 98.16 | 43.95 | 93.49 | 97.91 | 98.64 | 78.87 |
| **CIT (Ours)** | | **90.87** | 97.63 | 98.55 | 46.18 | 93.68 | 97.76 | 98.59 | 78.93 |
| **Triplet** | | 86.55 | 95.94 | 97.81 | 39.14 | 92.09 | 97.37 | 98.26 | 73.44 |
| **Angular** | | 86.94 | 96.74 | 98.13 | 39.75 | 91.74 | 97.05 | 97.87 | 72.78 |
| **CT** | **ViT** | 86.93 | 97.04 | 98.71 | 38.52 | 91.50 | 97.43 | 98.30 | 73.13 |
| **Circle** | | 81.55 | 94.35 | 96.68 | 36.67 | **92.73** | 97.66 | 98.32 | **76.97** |
| **ST** | | 87.09 | 96.83 | 98.61 | 36.96 | 91.10 | 97.02 | 98.00 | 71.12 |
| **CIT (Ours)** | | **88.54** | 96.32 | 98.00 | **43.57** | **92.73** | 98.03 | 98.67 | 76.62 |

parameter needed to be tuned carefully for ST, *i.e.*, the number of weight vectors per class. By contrast, our CIT avoids attentively regulating any hyper-parameters. The above three observations can demonstrate the elegance and effectiveness of our CIT, achieving comparable performance with the state-of-the-art and no need to tune hyper-parameters carefully.

## 3.3 IMAGE RETRIEVAL

We evaluate all comparable loss functions adopting default hyper-parameter settings on two image retrieval datasets, *i.e.*, Cars196 and In-shop Clothes. We compare our CIT against those state-of-the-art methods in Table 2. Given metrics $R@1$ and $mAP$ of two datasets on two networks, CIT gets four number one (bold) and four number two (underline) among all eight. Such results can again demonstrate the superiority of our CIT to other methods conditioned on no careful decision boundaries and other hyper-parameters tuning. There are other two interesting points implied in Table 2. First, for those comparable methods with the violation margin, Table 1 illustrates that various violation margins bring significant performance differences on the same dataset. Whereas Table 2 indicates that different datasets with the same violation margin exhibit significant gain differences. It can be proved by the performance of Circle which gains the best $R@1$ of 92.73 and $mAP$ of 76.97 on ViT for the In-shop Clothes dataset but achieves the worst on ViT for the Cars196 dataset. Second, for the Cars196 dataset, our CIT lay a lot behind in $mAP$ on CNN compared to Triplet. The metric $R@1$ only measures how many items are hit, while the metric $mAP$ also considers the rank of the hit items to reflect the ranking quality. From the standpoint of ranking quality, our CIT fails to quantize the concordance between the predicted and observed similarities. In other words, CIT implicitly models the relations of $S_{ap}$ and $S_{an}$ as inequality, while Triplet explicitly formulates their relations as an identical equation with the help of the violation margin. If the chosen violation margin for Triplet fits related tasks, such a prior constraint can help boost performance, and the obtained result is superior to our CIT. However, choosing an appropriate task-specific violation margin is challenging and laborious. Our CIT marginally outperforms its counterparts and exhibits an elegant training manner that does not need to adjust the violation margin elaborately.

## 3.4 FACE RECOGNITION

For the face recognition task, according to the ROC curve on TPR (True Positive Rate) at FPR (False Positive Rate) from 1e-6 to 1, we report AUC performances of four verification datasets on CNN and ViT in Figure 2. Our CIT gains the four highest AUC among the eight sub-tasks, including AgeDB on CNN (98.33), CFP-FP on CNN (97.39), IJB-C on ViT (99.32), and CFP-FP on ViT (96.85). The overall performance of this task can again affirm the availability of our CIT which directly optimizes the similarity concordance of triplet samples and is free from the annoyance of introducing the prior of the decision boundary. While regarding those loss functions

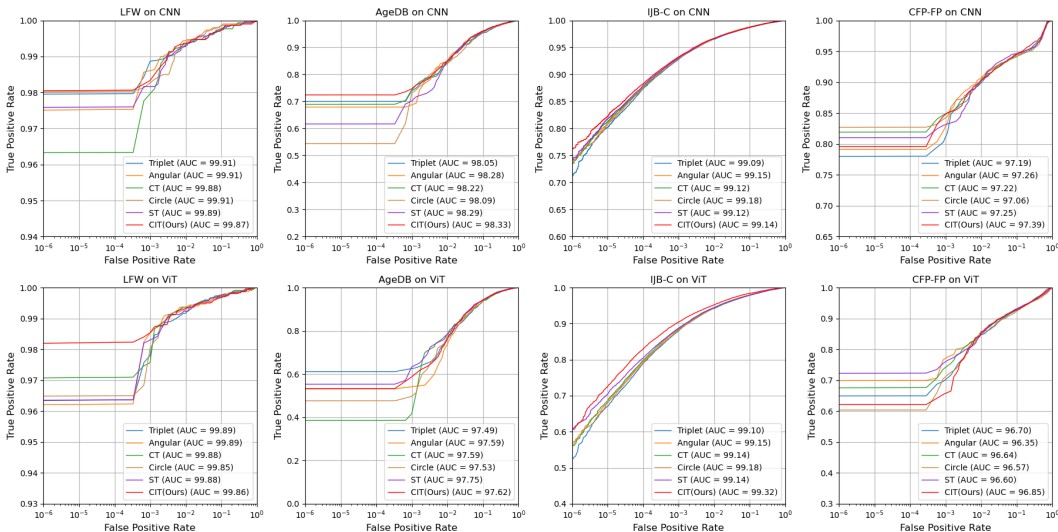

Figure 2: Comparison of different loss functions for LFW Huang et al. (2008), AgeDB Moschoglou et al. (2017), IJB-C Maze et al. (2018), and CFP-FP Sengupta et al. (2016) datasets on CNN (upper row) and ViT (down row) networks in terms of AUC (in %).

Table 3: Performance of our CIT loss function with different sizes of the hyper-parameter $\gamma$ on the Market1501 and In-shop Clothes datasets.

| $\gamma$ | Network | Market1501 | | | | In-shop Clothes | | | |
|---|---|---|---|---|---|---|---|---|---|
| | | $R@1$ | $R@5$ | $R@10$ | $mAP$ | $R@1$ | $R@5$ | $R@10$ | $mAP$ |
| 0.1 | | 95.72 | 98.69 | 99.23 | **89.51** | 93.14 | 97.76 | 98.43 | 78.68 |
| 0.2 | | 95.25 | 98.16 | 98.93 | 88.92 | **93.71** | 97.98 | 98.54 | 78.97 |
| 0.5 | CNN | 95.34 | 98.28 | 98.99 | 88.78 | 93.60 | 97.83 | 98.52 | **79.09** |
| 0.8 | | 95.34 | 98.52 | 99.05 | 88.88 | 93.60 | 97.78 | 98.51 | 78.89 |
| 1.0 | | **95.87** | 98.22 | 98.93 | 89.41 | 93.68 | 97.76 | 98.59 | 78.93 |
| 0.1 | | 93.74 | 97.86 | 99.02 | 86.09 | **92.76** | 98.02 | 98.40 | **76.64** |
| 0.2 | | 93.53 | 97.60 | 98.72 | 85.64 | 92.28 | 97.57 | 98.39 | 75.43 |
| 0.5 | ViT | 93.53 | 97.57 | 98.66 | 85.88 | 92.11 | 97.48 | 98.29 | 75.85 |
| 0.8 | | 93.71 | 97.74 | 98.75 | 86.06 | 92.21 | 97.66 | 98.45 | 75.41 |
| 1.0 | | **93.88** | 97.74 | 98.93 | **86.23** | 92.73 | 98.03 | 98.67 | 76.62 |

with the violation margin, we must diligently seek good violation margins for specific datasets and backbones to obtain comparable performance. And for ST without the violation margin, we still need to find an appropriate number of centers for each class, and the over-large number of centers raises an efficiency problem. Hyper-parameters are task-specific, so regulating hyper-parameters is indispensable and de-facto laborious and time-consuming. Based on the concordance relations of similarity, our CIT exhibits more flexibility in modeling triplet constraints than existing triplet-based losses. The violation margin enforces constant restrictions for triplet samples, while our CIT only abides unequal relationship. Without any bells and whistles on tuning hyper-parameters, CIT can yield advanced performances. And the only hyper-parameter $\gamma$ in CIT is task-agnostic.

## 3.5 ABLATION STUDY

Here we want to analyze the impact of the hyper-parameter $\gamma$ in Equation 8. Table 3 shows that the hyper-parameter $\gamma$ in our CIT loss has little effect on the performance. Both accuracy and mAP on different sizes of $\gamma$ exhibit consistency and coherence. For two datasets on CNN and ViT, the best of $R@1$ and $mAP$ (bold) scatter to varying sizes of $\gamma$. Since Equations 4 and 7 can be

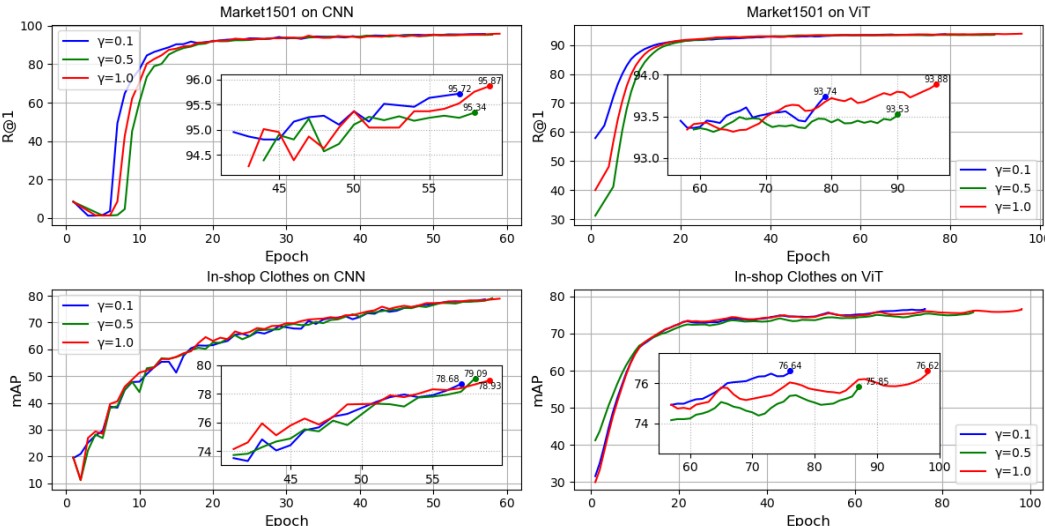

Figure 3: Rank-1 accuracy (R@1) (Market1501 dataset) and mAP (In-shop Clothes dataset) versus epochs of our CIT with different sizes of the hyper-parameter $\gamma$ on CNN and ViT.

interchangeable, and the difference is that Equation 7 lays more penalty on hard triplet samples. The reported stable performance can demonstrate that $\gamma$ has no impact on learning the concordance of triplet similarity in the mini-batch sampling. We can claim that this hyper-parameter is task-agnostic. From the performance perspective, our CIT can pay no attention to this only hyper-parameter. On the other hand, from the convergence viewpoint, introducing $\gamma$ aims to help DML models speed up the convergence by laying emphasis on hard triplet samples. As Figure 3 shows, CIT with lesser $\gamma$ can be fast convergence with fewer epochs achieving the last performance. The lesser $\gamma$ entails that Equation 7 with more loose concordance contributes more to the loss value. Our CIT pays more penalty on hard triplet samples and little penalty on easy triplet samples by setting lesser $\gamma$, thus advancing the convergence speed. This ablation study can demonstrate that our CIT with $\gamma$ can alleviate the training complexity of triplet-based loss functions.

## 4 CONCLUSIONS AND DISCUSSIONS

Building on the concordance constraint of triplet similarity, we propose a novel and elegant concordance-induced triplet (CIT) loss function to simplify the optimization process for deep metric learning (DML). Our CIT loss can free DML training from the laborious tuning of the violation margin in conventional triplet-based loss functions and encourage model training fast convergence. The violation margin is task-specific, while the concordance constraint is task-agnostic and monotonous. Hence, the concordance between the predicted and observed similarities can help our CIT loss push far away from the plague of giving prior constraints for decision boundaries. We further utilize the degree of concordance of triplet samples to pay more penalties on hard triplet samples to speed up gradient optimization. The extensive experiments on three popular DML tasks with two networks can demonstrate the elegance and availability of our proposed CIT, yielding performances on par with other triplet-based loss functions.

It is worthy to emphasize that our CIT loss intends to achieve comparable performance with its counterparts but not pursue superior performance. CIT can favor DML training simply and elegantly by modeling the concordance of triplet similarity. When a DML task is challenging to offer the violation margin and needs to alleviate the training complexity, our CIT loss is a reliable alternative to conventional triplet-based loss functions. It is interesting to explore the degree of concordance of triplet samples to bring the best performance in the future. If we can design a method to measure the degree of concordance, we can avoid the excessive constraints of the partial likelihood term in our CIT loss.

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

## A  GENERALIZATION BOUND ANALYSIS

**Theorem 1.** Let $\mathbb{L}$ represents a family of loss function associated to $\mathcal{L}_{cit}$ given DML model $\mathcal{F}_{\boldsymbol{\theta}}$. Then for $\forall \delta > 0$ over $\mathrm{E}_{\mathcal{T}}$, each of the following holds for any $\mathcal{L}_{cit} \in \mathbb{L}$ with probability at least $1 - \delta$.

$$\mathbb{E}[\mathcal{L}_{cit}] \leq \tilde{\mathbb{E}}_{\mathcal{D}}[\mathcal{L}_{cit}] + R_{\mathcal{D}}(\mathbb{L}) + \sqrt{\frac{\ln(1/\delta)}{2N_{\mathcal{T}}}}, \tag{9}$$

where $\tilde{\mathbb{E}}_{\mathcal{D}}[\mathcal{L}_{cit}]$ is the empirical error $\mathcal{G}(\mathcal{F}_{\boldsymbol{\theta}})$ in Equation 5 over the training set $\mathcal{D}$ (in-of-samples), and $\mathbb{E}[\mathcal{L}_{cit}]$ denotes the generalization or expectation error on out-of-samples. $\mathrm{E}_{\mathcal{T}}$ with length $N_{\mathcal{T}}$ is sampled from the training set $\mathcal{D}$. $R_{\mathcal{D}}(\mathbb{L})$ signifies the Rademacher complexity of $\mathbb{L}$ with respect to the training set $\mathcal{D}$.

**Proof Sketch.** The generalization error bound based on McDiarmid inequality and Rademacher complexity for concordance learning can be proved by Theorem 3.5 described in Mohri et al. (2018).

**Remark 1.** As shown in Equation 9, the supremum of the generalization bound consists of three terms. The first term is the empirical error relating to training, the lower empirical error brings a smaller supremum of the generalization bound. And in the third term, the more considerable amount of triplet samples $\mathcal{T}$ can reduce the upper bound of generalization error. The generalization bound of our loss function $\mathcal{L}_{cit}$ given the DML model $\mathcal{F}_{\boldsymbol{\theta}}$ is largely determined by the second term, Rademacher complexity of $\mathbb{L}$.

**Theorem 2.** For the hypothesis space $\mathbb{L} = \{\mathbb{L} : \mathcal{T} \to \{0, 1\}\}$, we state $\mathcal{L}_{cit}(\mathcal{T}_1, \mathcal{T}_2, \cdot, \mathcal{T}_N) = \left(\mathcal{L}_{cit}(\mathcal{T}_1), \mathcal{L}_{cit}(\mathcal{T}_2), \cdots, \mathcal{L}_{cit}(\mathcal{T}_N)\right) \in \{0, 1\}^N$ is a dichotomy. We further define the growth function of $\mathbb{L}$ is $M_{\mathbb{L}}(N_{\mathcal{T}}) = \max_{N_{\mathcal{T}}} \left| \mathbb{L}(\mathcal{T}_1, \mathcal{T}_2, \cdots, \mathcal{T}_N) \right|$. Then the following holds:

$$R_{\mathcal{D}}(\mathbb{L}) \leq \sqrt{\frac{2 \ln M_{\mathbb{L}}(N_{\mathcal{T}})}{N_{\mathcal{T}}}}. \tag{10}$$

**Proof Sketch.** Relating the Rademacher complexity to the growth function, Equation 10 can be derived by Theorem 3.7 (Massart's lemma) and Corollary 3.8 in Mohri et al. (2018).

**Remark 2.** The Rademacher complexity can be bounded in terms of the growth function, which is distribution independent and purely combinatorial. The growth function $M_{\mathbb{L}}(N_{\mathcal{T}})$ is the maximum number of distinct ways in which triplet samples of $N_{\mathcal{T}}$ can be predicted as concordance (0) or discordance (1) by using hypotheses in $\mathbb{L}$. It suggests the number of dichotomies realized by the

hypothesis and its upper bound is $2^{N_\mathcal{T}}$. The growth function $M_\mathbb{L}(N_\mathcal{T})$ suggests the representation power of the hypothesis space $\mathbb{L}$, thus reflecting the complexity of the hypothesis space.

Combining Equations 9 and 10, we rewrite the supremum of the generalization error as:

$$\mathbb{E}[\mathcal{L}_{cit}] = \sup_\mathcal{D} \left\{ \tilde{\mathbb{E}}_\mathcal{D}[\mathcal{L}_{cit}] + M_\mathbb{L}(N_\mathcal{T}) + \sqrt{\frac{\ln(1/\delta)}{2N_\mathcal{T}}} \right\}. \tag{11}$$

Based on the above theorems, we would like to compare the supremum of the generalization bound of two items in the loss function $\mathcal{L}_{cit}$. The relation of triplets is modeled as $S_{ap} - S_{an}$ in Equation 4, and we reorganize the modeling relation of Equation 7 with a violation margin, as follows:

$$S_{ap} - S_{an} + \underbrace{\left( S_{an} - \log(e^{S_{an}} + e^{S_{pn}}) \right)}_{margin}. \tag{12}$$

Similar to the standard triplet loss in Equation 2, the loss function $\mathcal{L}_p$ utilizes a margin to constrain triplet similarities, as Equation 12 shows. Obviously, from the standpoint of modeling complexity, the representation power of $\mathcal{L}_p$ is stronger than $\mathcal{L}_e$.

**Corollary 1.** Given the same DML model $\mathcal{F}_\theta$, we let $\mathbb{L}_e$ represents a family of loss function associated to $\mathcal{L}_e$ and $\mathbb{L}_p$ represents a family of loss function associated to $\mathcal{L}_p$. Then the following holds:

$$M_{\mathbb{L}_e}(N_\mathcal{T}) < M_{\mathbb{L}_p}(N_\mathcal{T}) \leq 2^{N_\mathcal{T}}. \tag{13}$$

For any hypothesis set $\mathbb{L}$, the trivial bound is $2^{N_\mathcal{T}}$. The growth function measures the richness or complexity of the hypothesis set $\mathbb{L}$. Hence, the growth function $M_{\mathbb{L}_e}(N_\mathcal{T})$ for the hypothesis set $\mathbb{L}_e$ is smaller than the growth function $M_{\mathbb{L}_p}(N_\mathcal{T})$ for the hypothesis set $\mathbb{L}_p$.

**Corollary 2.** By virtue of **Theorem 1** and **Theorem 2**, we further bound the difference between the empirical error and generalization error for loss functions $\mathbb{L}_e$ and $\mathbb{L}_p$ associated to the DML model $\mathcal{F}_\theta$.

$$\mathbb{E}[\mathcal{L}_e] - \tilde{\mathbb{E}}_\mathcal{D}[\mathcal{L}_e] \leq M_{\mathbb{L}_e}(N_\mathcal{T}) + \sqrt{\frac{\ln(1/\delta)}{2N_\mathcal{T}}}. \tag{14}$$

$$\mathbb{E}[\mathcal{L}_p] - \tilde{\mathbb{E}}_\mathcal{D}[\mathcal{L}_p] \leq M_{\mathbb{L}_p}(N_\mathcal{T}) + \sqrt{\frac{\ln(1/\delta)}{2N_\mathcal{T}}}. \tag{15}$$

With Equation 13, we can obtain:

$$\mathbb{E}[\mathcal{L}_e] - \tilde{\mathbb{E}}_\mathcal{D}[\mathcal{L}_e] \leq \mathbb{E}[\mathcal{L}_p] - \tilde{\mathbb{E}}_\mathcal{D}[\mathcal{L}_p]. \tag{16}$$

Supposing the amount of triplet samples is enough to make both expectation errors theoretically infinite close. Then, we can further derive $\tilde{\mathbb{E}}_\mathcal{D}[\mathcal{L}_p] \leq \tilde{\mathbb{E}}_\mathcal{D}[\mathcal{L}_e]$.

**Remark 3.** Due to $\mathcal{L}_p$ introducing an extra decision boundary, the complexity of modeling triplet similarities can better fit the training set, thus helping speed training convergence. Compared to the loose constraint of $\mathcal{L}_p$, $\mathcal{L}_e$ enforces a tight constraint on triplet similarities. As shown in in (b) of Figure 1, our loss function $\mathcal{L}_{cit}$ fulfills lower loss by setting $\gamma = 0.5$.

However, we need to make a trade-off between the training error and generalization error on the complexity of the hypothesis space. The larger complexity of the hypothesis space easily leads to higher generalization errors. Such as, one of the popular strategies for controlling the complexity to avoid over-fitting is introducing regularization terms. Therefore, it is not that the higher the complexity and the smaller the training error can surely bring the better generalization.

Table 3 can account for the necessity of holding the trade-off between the training error and generalization error. Increasing complexity by setting $\gamma < 1.0$, *i.e.*, introducing $\mathcal{L}_p$ in $\mathcal{L}_{cit}$ favors the In-shop Clothes dataset with 11,735 classes. On the contrary, the best performances of the Market1501 dataset containing 1,501 identities are achieved by setting $\gamma = 1.0$, *i.e.*, without $\mathcal{L}_p$.

Through generalization bound and experiment result analysis, we can conclude:

- To free DML training from tuning the decision boundary, we present a tight constraint on triplet similarities and achieve comparable performances with its counterparts.

- We further introduce a loose strategy to increase the complexity of the hypothesis space to speed convergence.

- Increasing complexity is favorable to a large amount of training set. When the number of triplet samples is enough, we can regulate a relatively small $\gamma$ to enjoy two benefits simultaneously: speeding convergence and better generalization.

## B  GRADIENTS ON SIMILARITIES OF COMPARABLE PAIRS

The partial derivatives of our CIT loss in Equation 8 with respect to $S_{ap}$ and $S_{an}$ can be written as Equations 17 and 18, and the schematics of both corresponding gradients are shown in Figure 4.

$$\frac{\partial \mathcal{L}_{cit}}{\partial S_{an}} = \gamma e^{-(S_{an}-S_{ap})} - (1-\gamma)\frac{e^{S_{an}}}{(e^{S_{an}}+e^{S_{pn}})\ln}. \qquad (17)$$

$$\frac{\partial \mathcal{L}_{cit}}{\partial S_{ap}} = -\gamma e^{-(S_{an}-S_{ap})} + (1-\gamma), \qquad (18)$$

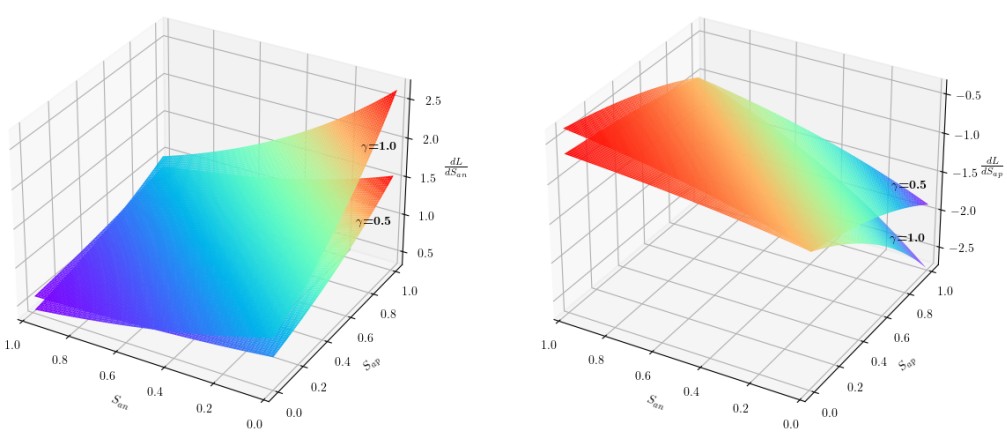

Figure 4: Gradients of CIT Loss Function with respect to $S_{ap}$ and $S_{an}$. CIT loss lays emphasize on hard samples by setting $\gamma = 0.5$ compared to $\gamma = 1.0$.

## C  PERFORMANCE OF FACE RECOGNITION

In Figure 2, we show the full ROC curves for comparable loss functions. Here, we additionally report their true accepted rate (TAR) at 1e-5 and 1e-3 false accepted rate (FAR) in Tables 4 and 5. Besides, we also compare their face verification accuracy in Table 6.

## D  EMBEDDING SPACE METRIC

An image can be encoded into an embedding feature by a differentiable DML model trained by a triplet loss function. In Tables 7 and 8, we report performances for Triplet and CT on Market1501 and In-shop Clothes datasets by tuning more margins. Besides, we also provide two metrics to measure embedding space, *i.e.*, normalized mutual information (NMI), spectral variance (SV) Roth et al. (2020). Lower values of SV indicate more directions of significant variance and suggest more discriminative embedding features.

Table 4: Comparison of different loss functions for LFW Huang et al. (2008), AgeDB Moschoglou et al. (2017), IJB-C Maze et al. (2018), and CFP-FP Sengupta et al. (2016) datasets on CNN in terms of TAR@FAR (in %).

| Method | LFW | | AgeDB | | IJB-C | | CFP-FP | |
|---|---|---|---|---|---|---|---|---|
| @CNN | TAR@FAR | | TAR@FAR | | TAR@FAR | | TAR@FAR | |
| | 1e-5 | 1e-3 | 1e-5 | 1e-3 | 1e-5 | 1e-3 | 1e-5 | 1e-3 |
| Triplet | 92.73 | **98.87** | 64.07 | 74.00 | 80.20 | 92.77 | 67.89 | 80.23 |
| Angular | 88.07 | 98.57 | 65.30 | 67.93 | 80.80 | 93.05 | 77.54 | 82.03 |
| CT | **95.37** | 97.97 | 53.77 | 74.33 | 81.24 | 93.04 | 72.97 | **84.69** |
| Circle | 91.43 | 98.27 | 51.27 | 73.13 | 80.82 | 92.93 | 79.03 | 83.97 |
| ST | 92.57 | 98.17 | 56.30 | 70.20 | 81.75 | 93.08 | **80.37** | 83.00 |
| CIT (Ours) | 91.67 | 98.33 | **71.37** | **74.70** | **82.23** | **93.24** | 76.09 | 84.46 |

Table 5: Comparison of different loss functions for LFW Huang et al. (2008), AgeDB Moschoglou et al. (2017), IJB-C Maze et al. (2018), and CFP-FP Sengupta et al. (2016) datasets on ViT in terms of TAR@FAR (in %).

| Method | LFW | | AgeDB | | IJB-C | | CFP-FP | |
|---|---|---|---|---|---|---|---|---|
| @ViT | TAR@FAR | | TAR@FAR | | TAR@FAR | | TAR@FAR | |
| | 1e-5 | 1e-3 | 1e-5 | 1e-3 | 1e-5 | 1e-3 | 1e-5 | 1e-3 |
| Triplet | 87.13 | 97.80 | **56.63** | 62.90 | 67.01 | 87.96 | 54.94 | 69.63 |
| Angular | 86.00 | **98.53** | 40.30 | 53.17 | 68.02 | 88.43 | 49.54 | **76.57** |
| CT | 87.07 | 97.57 | 35.10 | 41.67 | 68.49 | 88.62 | 39.06 | 73.54 |
| Circle | 91.17 | 98.03 | 42.83 | 49.70 | 68.10 | 88.23 | 59.29 | 70.43 |
| ST | **91.50** | 98.30 | 50.50 | **63.33** | 70.40 | 88.74 | 42.00 | 75.46 |
| CIT (Ours) | 83.77 | **98.53** | 51.93 | 58.70 | **72.45** | **90.43** | 60.89 | 65.40 |

Table 6: Comparison of different loss functions for LFW Huang et al. (2008), AgeDB Moschoglou et al. (2017), IJB-C Maze et al. (2018), and CFP-FP Sengupta et al. (2016) datasets on CNN and ViT networks in terms of face verification accuracy (in %).

| Method | CNN | | | | ViT | | | |
|---|---|---|---|---|---|---|---|---|
| | LFW | AgeDB | IJB-C | CFP-FP | LFW | AgeDB | IJB-C | CFP-FP |
| Triplet | 99.37 | 93.97 | 98.23 | 94.93 | 99.27 | 92.12 | 98.25 | 92.90 |
| Angular | 99.10 | 93.65 | 98.27 | 94.80 | 99.07 | 91.50 | 98.24 | 92.30 |
| CT | 99.30 | 94.25 | **98.52** | 94.83 | **99.40** | 92.77 | 98.28 | **93.39** |
| Circle | 99.35 | 94.22 | 98.32 | 93.99 | 99.26 | 92.63 | 98.35 | 93.19 |
| ST | 99.37 | 94.52 | 98.28 | **95.19** | 99.35 | 93.07 | 97.99 | 93.27 |
| CIT (Ours) | **99.40** | **94.57** | 98.40 | 94.76 | 99.37 | **93.18** | **98.39** | 93.34 |

## E  CONVERGENCE CURVES

Figure 5 and 6 help analyze the convergence of loss functions.

## F  QUALITATIVE RESULTS

In Figure 7, we exhibit some retrieval cases for our CIT loss on different datasets.

## G  GRADIENTS ON FEATURES OF TRIPLET SAMPLES

In Figure 8, we visualize the gradients of three features $x_a$, $x_p$, $x_n$. The three features may belong to different classes under randomly mini-batch sampling, thus incomparable. Hence, from this visualization, we can not discern the contributions of features to DML optimization.

Table 7: Comparison of Triplet and CT with different margins for the Market1501 dataset on CNN and ViT networks in terms of rank-1 (in %) accuracy, normalized mutual information (NMI, in %), spectral variance (SV, in %), mean average precision (mAP, in %), and mean inverse negative penalty (mINP, in %).

| Method | Margin | Network | Market1501 | | | | |
|---|---|---|---|---|---|---|---|
| | | | $R@1$ | $mAP$ | $mINP$ | $SV$ | $NMI$ |
| **Triplet** | $m = 0.0$ | | 95.64 | 88.73 | 66.60 | 44.37 | 95.59 |
| | $m = 0.01$ | | 95.58 | 89.08 | 67.05 | 44.45 | 95.56 |
| | $m = 0.05$ | | 95.75 | 89.26 | 67.48 | 44.44 | 95.69 |
| | $m = 0.1$ | | 95.75 | 89.19 | 66.65 | 43.83 | 95.50 |
| | $m = 0.5$ | | 95.40 | 88.78 | 66.46 | 45.86 | 95.57 |
| | $m = 1.0$ | | 95.37 | 88.73 | 66.64 | 49.08 | 95.62 |
| **CT** | $m = 0.0$ | **CNN** | 95.81 | 88.77 | 66.32 | 44.02 | 95.54 |
| | $m = 0.01$ | | 95.52 | 88.68 | 66.11 | 44.54 | 95.55 |
| | $m = 0.05$ | | 95.61 | 89.05 | 66.28 | 44.84 | 95.61 |
| | $m = 0.1$ | | 95.16 | 88.65 | 66.27 | 46.90 | 95.51 |
| | $m = 0.5$ | | 95.43 | 88.79 | 67.07 | 44.81 | 95.51 |
| | $m = 1.0$ | | 95.64 | 88.74 | 66.24 | 44.10 | 95.65 |
| **CIT (Ours)** | - | | **95.87** | **89.41** | **68.27** | **43.62** | **95.70** |
| **Triplet** | $m = 0.0$ | | 93.56 | 84.76 | 57.45 | 38.00 | 95.15 |
| | $m = 0.01$ | | 93.62 | 85.07 | 58.88 | 38.18 | 95.19 |
| | $m = 0.05$ | | **94.06** | 85.87 | 60.74 | **36.25** | 95.32 |
| | $m = 0.1$ | | 93.68 | 86.02 | **61.94** | 36.96 | 95.19 |
| | $m = 0.5$ | | 93.82 | 84.76 | 60.60 | 36.86 | 95.27 |
| | $m = 1.0$ | | 93.68 | 84.64 | 58.81 | 37.95 | 95.20 |
| **CT** | $m = 0.0$ | **ViT** | 92.37 | 81.79 | 51.82 | 51.79 | 95.01 |
| | $m = 0.01$ | | 92.19 | 81.75 | 51.91 | 54.50 | 95.08 |
| | $m = 0.05$ | | 92.58 | 82.27 | 52.87 | 47.84 | 95.11 |
| | $m = 0.1$ | | 93.38 | 83.00 | 54.31 | 38.24 | 95.06 |
| | $m = 0.5$ | | 92.99 | 84.62 | 60.03 | 48.71 | 95.31 |
| | $m = 1.0$ | | 93.47 | 84.16 | 58.44 | 46.43 | 95.22 |
| **CIT (Ours)** | - | | 93.88 | **86.23** | 61.88 | 36.65 | **95.38** |

Table 8: Comparison of Triplet and CT with different margins for the In-shop Clothes dataset on CNN and ViT networks in terms of rank-1 (in %) accuracy, normalized mutual information (NMI, in %), spectral variance (SV, in %), mean average precision (mAP, in %), and mean inverse negative penalty (mINP, in %).

| Method | Margin | Network | In-shop Clothes | | | | |
|---|---|---|---|---|---|---|---|
| | | | $R@1$ | $mAP$ | $mINP$ | $SV$ | $NMI$ |
| **Triplet** | $m = 0.0$ | | 93.53 | 78.21 | 61.71 | 47.99 | 94.48 |
| | $m = 0.01$ | | 93.44 | 78.60 | 62.34 | 47.83 | 94.47 |
| | $m = 0.05$ | | 93.37 | 78.47 | 62.58 | 47.91 | 94.54 |
| | $m = 0.1$ | | 93.66 | 78.36 | 62.09 | 47.63 | 94.51 |
| | $m = 0.5$ | | 93.39 | 78.61 | 62.40 | 48.85 | 94.54 |
| | $m = 1.0$ | | 93.37 | 78.07 | 61.57 | 50.31 | 94.39 |
| **CT** | $m = 0.0$ | **CNN** | 93.44 | 78.73 | 62.37 | 48.72 | 94.53 |
| | $m = 0.01$ | | 93.37 | 78.33 | 61.88 | 50.45 | 94.45 |
| | $m = 0.05$ | | **93.75** | **79.17** | 62.87 | **46.64** | **94.62** |
| | $m = 0.1$ | | 93.44 | 78.27 | 61.87 | 48.33 | 94.51 |
| | $m = 0.5$ | | 93.57 | 78.33 | 61.92 | 48.31 | 94.46 |
| | $m = 1.0$ | | 93.61 | 78.43 | 62.12 | 48.09 | 94.49 |
| **CIT (Ours)** | - | | 93.68 | 78.93 | **62.92** | 46.76 | 94.58 |
| **Triplet** | $m = 0.0$ | | 92.31 | 74.84 | 56.50 | 40.00 | 93.59 |
| | $m = 0.01$ | | 92.52 | 75.97 | 57.95 | 37.81 | 93.93 |
| | $m = 0.05$ | | 92.09 | 73.44 | 55.62 | 39.66 | 93.81 |
| | $m = 0.1$ | | 92.70 | 76.58 | 59.19 | 37.62 | 93.93 |
| | $m = 0.5$ | | 92.55 | 75.84 | 58.64 | 38.99 | 93.85 |
| | $m = 1.0$ | | 92.45 | 74.50 | 56.82 | 38.33 | 93.59 |
| **CT** | $m = 0.0$ | **ViT** | 91.35 | 72.43 | 53.40 | 39.54 | 93.12 |
| | $m = 0.01$ | | 91.49 | 72.63 | 53.63 | 38.99 | 93.19 |
| | $m = 0.05$ | | 91.50 | 73.13 | 54.40 | 38.49 | 93.30 |
| | $m = 0.1$ | | 91.67 | 74.08 | 55.52 | 38.40 | 93.50 |
| | $m = 0.5$ | | 92.64 | 75.43 | 58.01 | 38.16 | 93.75 |
| | $m = 1.0$ | | 92.47 | 74.65 | 56.91 | 38.37 | 93.64 |
| **CIT (Ours)** | - | | **92.73** | **76.62** | **59.83** | **37.51** | **93.94** |

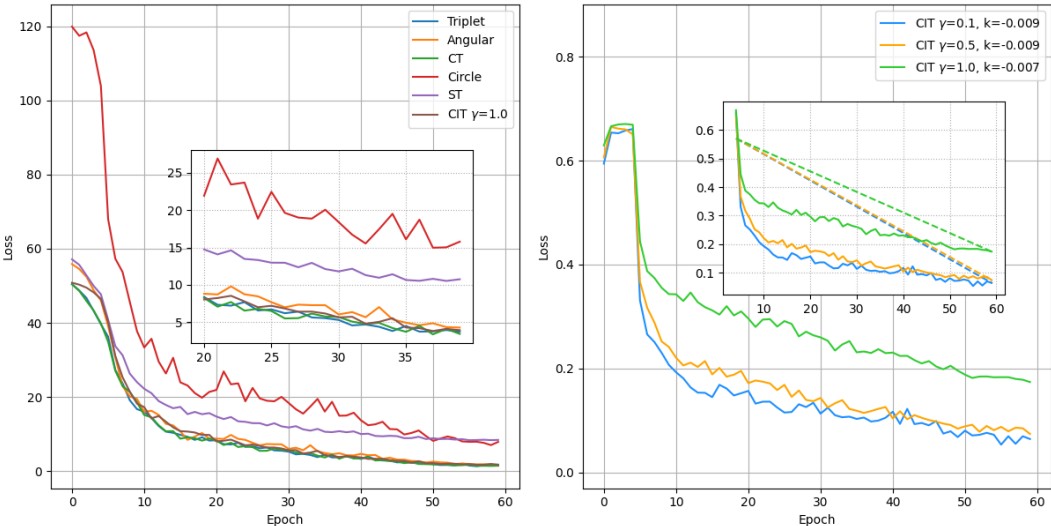

Figure 5: Convergence curves of CNN for different loss functions on the Market1501 dataset. Left shows that our CIT can achieve the same convergence effect as Triplet and CT, smoother than Circle, ST, and Angular, which need tuning more hyper-parameters. And right plot again proves that the hyper-parameter can help speed the convergence of our CIT. $k$ signifies the angle factor of a straight line and is used to measure the decline speed of loss curves roughly.

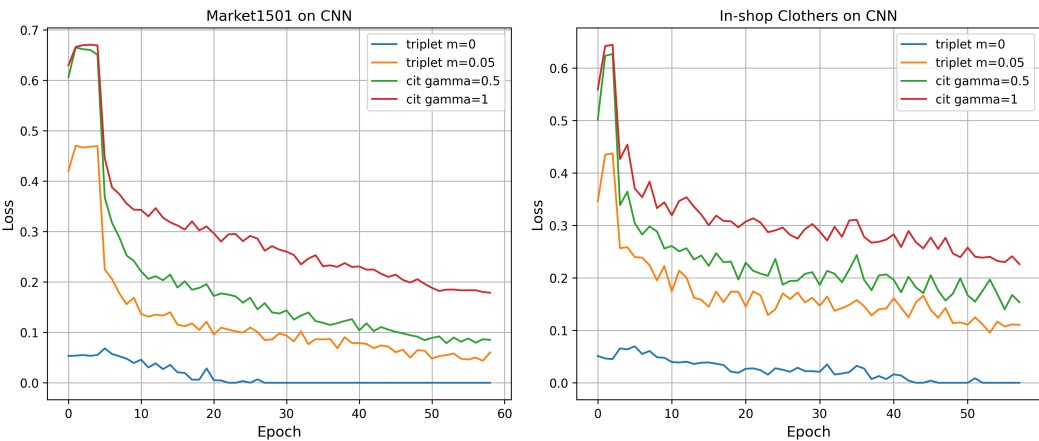

Figure 6: Convergence curves of CNN for our CIT and Triplet loss functions on CNN for the Market1501 and In-shop Clothes dataset. Triplet of $m = 0.0$ falls into the local optimum prematurely by observing occurrence epochs of the zero loss. Triplet of m=0.0 has the same mining strategy as our CIT, but its $L_1$ form is weaker than CIT in the exponential form in driving DML model optimization due to insufficient exploitation of hard triplet samples. By setting $\gamma < 1.0$, our CIT building on similarity concordance can approach the convergence of Triplet with m=0.05.

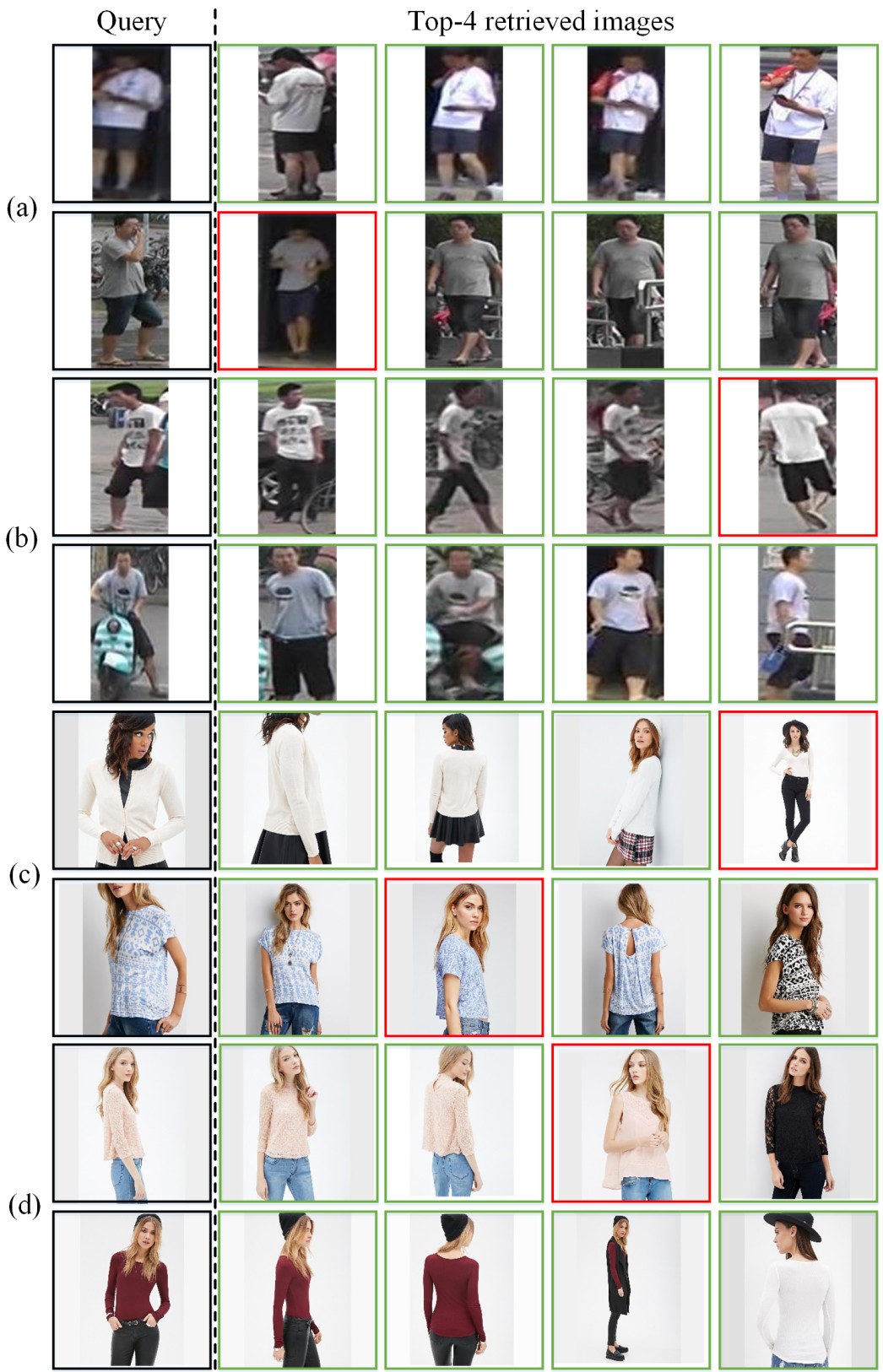

Figure 7: Qualitative results of our CIT loss for (a) Market1501 dataset on CNN, (b) Market1501 dataset on ViT, (c) In-shop Clothes dataset on CNN, and (d) In-shop Clothes dataset on ViT. For each query image (leftmost), top 4 retrievals are exhibited. The results with red boundaries are false cases but they are substantially similar to the query images in terms of appearance.

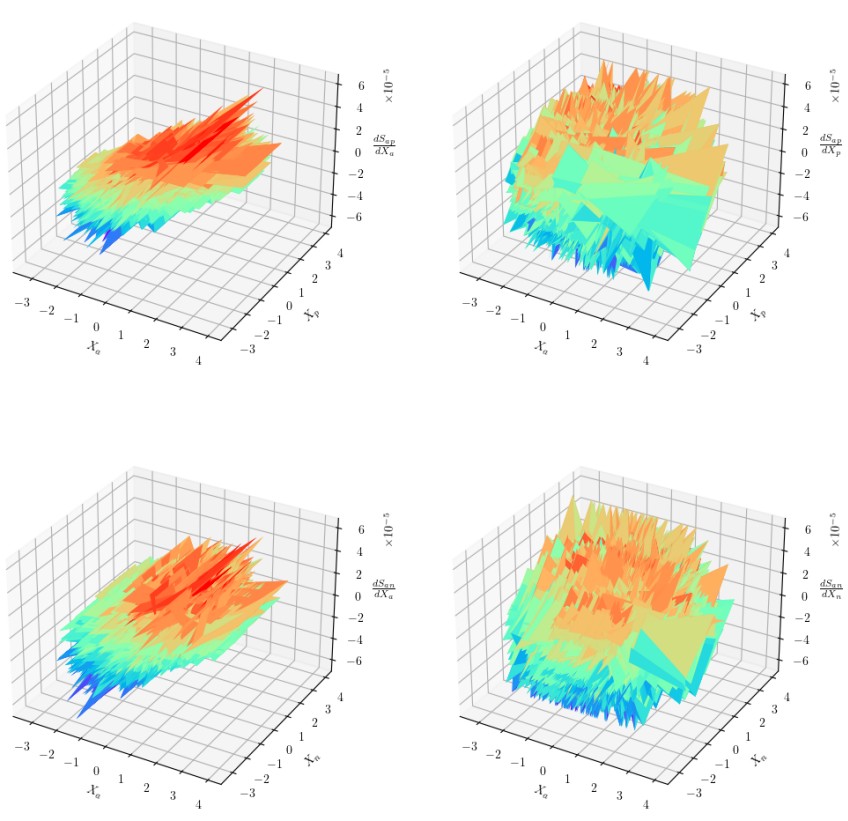

Figure 8: Gradients on features of triplet samples.

