# OpenReview forum: "Triplet Similarity Learning on Concordance Constraint"
_ICLR.cc/2023/Conference — Submitted to ICLR 2023_

### Official Review · Reviewer_nK2W · 2022-10-19

**Confidence:** 5
**Correctness:** 3
**Technical Novelty And Significance:** 2
**Empirical Novelty And Significance:** 2
**Recommendation:** 3

**Clarity, Quality, Novelty And Reproducibility:**

The paper is well-written and clear to read
The novelty is incremental due to the introduction of the additional hyper-parameter.

**Strength And Weaknesses:**

Strength
The proposed CIT triplet loss avoids tuning the margin hyper-parameter used in the regular triplet loss.
The experiments are well-conducted across different types of datasets
The paper is well-organized and written.

Weaknesses
Though multiple datasets results are provided, they are all about retrieval performance. There should be other types of evidence to support the improvement of the new loss functions such as the embedding space structure(TSNE plots), embedding space density, and spectral decay[1] which helps understanding and metrics to show where is gained coming from with the new loss function.

The author proposes to avoid the tuning of the margin hyper-parameter. But the new loss function introduces another hyper-parameter \gama, which is dimming the novelty.

In figure 3, why all the ending points are not with the same epoch size? The training of VIT on In-shop Clothes is not stable. Suggestion: it's better to train the model multiple times and average the curve for these plots.

it's much more interesting to see the gradient difference for the CIT loss and have more discussion on the gradient level and whether the proposed idea fits the explanation of the gradient. In the appendix, it's better to show the gradient in the format of the loss term w.r.t to features x_a, x_p,x_n


some minor issues
in Section 2.1 the similar hard triplet samples definition should be written in Sap-m<San<Sap. Although it's the same equation, semi-hard mostly describes the negative example. Put San in the relation to Sap is clear.
missing the definition of Spn


[1] Karsten Roth, Timo Milbich, Samarth Sinha, Prateek Gupta, Björn Ommer, and Joseph Paul Cohen.
Revisiting training strategies and generalization performance in deep metric learning. In Proceedings of the
37th International Conference on Machine Learning, ICML 2020, 13-18 July 2020, Virtual Event, volume
119 of Proceedings of Machine Learning Research, pages 8242–8252, 2020.

**Summary Of The Paper:**

The paper proposed a Concordance-induced triplet(CIT) loss for Deep Metric Learning tasks. The major hypothesis is that the ordering concordance should be invariant to any monotone transformation of the decision boundary of triplet loss. Therefore, CIT loss should with concordance can help avoid the plague of turning the violation margin. Moreover, the paper also introduces the partial likelihood term on hard triplets to speed up convergence. Multiple benchmark experiments are conducted for the new loss.

**Summary Of The Review:**

In sum, SOTA is important for DML research but it's not the only type of metric to evaluate the new loss function. Please check out the Strength And Weaknesses.

---

> ### Author Response · Authors · 2022-11-14
> **Response to Reviewer nK2W**
>
> $\textbf{Q1}$: Though multiple datasets results are provided, they are all about retrieval performance. There should be other types of evidence to support the improvement of the new loss functions such as the embedding space structure(TSNE plots), embedding space density, and spectral decay\cite{roth2020revisiting} which helps understanding and metrics to show where is gained coming from with the new loss function.
>
> $\textbf{A1}$: According to your insightful suggestions, we report normalized mutual information (NMI) and spectral variance (SV) [1] in Tables 7 and 8 to explain the learned embedding space. Our CIT achieves consistent performances with retrieval metrics in terms of NMI for evaluating clustering. Similarly, SV and R@1 are consistent in performance, and our CIT achieves the two lowest SVs among the four tasks. The lower SV suggests better generalization. The explanation of SV and NMI again proves that our CIT can help DML models learn distinguishing embedding spaces, thus generating comparable retrieval performances with state-of-the-art.
>
> $\textbf{Q2}$: The author proposes to avoid the tuning of the margin hyper-parameter. But the new loss function introduces another hyper-parameter $\gamma$, which is dimming the novelty.
>
> $\textbf{A2}$: In our CIT loss, the hyper-parameter $\gamma$ is task-agnostic, while the decision margin in conventional triplet losses is task-specific. Enhancing the exploitation of hard triplet samples is beneficial to training convergence. Hence, we introduce a partial likelihood term $\mathcal{L}_{p}$ by setting $\gamma<1.0$ in our CIT. This term aims to enhance penalty on hard triplet samples to help speed up convergence, and it plays an insignificant effect on the final performance. In the ablation study, the experimental results of Table 3 can prove this claim. Hence, it is not necessary to laboriously tune its hyper-parameter $\gamma$. In our three experiments, $\gamma$ always is 1.0, i.e., the partial likelihood term is disabled.
>
> $\textbf{Q3}$: In figure 3, why all the ending points are not with the same epoch size? The training of VIT on In-shop Clothes is not stable. Suggestion: it's better to train the model multiple times and average the curve for these plots.
>
> $\textbf{A3}$: With the same epoch, Figure 3 shows that $\gamma<1.0$ can reach its highest performances in advance than $\gamma=1.0$. To display such a comparison, we use the epoch where the highest point lies as the ending point. This can explain the effect of setting $\gamma$ on convergence speed. According to your valuable advice, we re-plot curves in Figure 3 on the average values based on multiple ViT training.
>
> $\textbf{Q4}$: some minor issues in Section 2.1, the similar hard triplet samples definition should be written in Sap-m<San<Sap. Although it's the same equation, semi-hard mostly describes the negative example. Put San in relation to Sap is clear. missing the definition of Spn.
>
> $\textbf{A4}$: Thanks for your careful reviews. We have rewritten the relation for semi-hard triplet samples in Sec.2.2 and added the definition for $S_{pn}$.
>
> $\textbf{Q5}$: It's much more interesting to see the gradient difference for the CIT loss and have more discussion on the gradient level and whether the proposed idea fits the explanation of the gradient. In the appendix, it's better to show the gradient in the format of the loss term w.r.t to features $x_{a}$, $x_{p}$,$x_{n}$.
>
> $\textbf{A5}$: In Figure 4, we visualize the gradient surfaces of our CIT loss on $\gamma=1.0$ and $\gamma=0.5$ respectively w.r.t two similarities $S_{ap}$ and $S_{an}$. This visualization aims to reflect the contributions to DML optimization of imposing more penalties on hard triplet samples by setting $\gamma<1.0$. The similarity pair $S_{ap}$ and $S_{an}$ is comparable. $S_{ap}<S_{an}$ suggests a hard triplet samples. Hence, it is significant to explore their gradient difference. Differently, three features $x_{a}$, $x_{p}$,$x_{n}$  may belong to different classes under mini-batch sampling and thus are incomparable. We can not effectively observe their contributions to gradients, as Figure 8 shown.
>
> [1]Roth, Karsten, et al. "Revisiting training strategies and generalization performance in deep metric learning." International Conference on Machine Learning. PMLR, 2020.

---

### Official Review · Reviewer_mdeV · 2022-10-24

**Confidence:** 4
**Correctness:** 3
**Technical Novelty And Significance:** 3
**Empirical Novelty And Significance:** 2
**Recommendation:** 3

**Clarity, Quality, Novelty And Reproducibility:**

**Clarity**. The paper is quite well written despite the typo pointed out in the weakness #3 above.

**Quality**. The proposed method is technically sounding, and the experiments more of less validate the claims. But as elaborated above, it lacks definition on the key concept "concordance", lacks the discussions on detailed technical choices, and the significance of the results is less convincing, which dampens the overall quality.

**Novelty**. The proposed method can be considered a novel contribution, although the significance is arguably marginal, given that it is a variant based on the conventional triplet loss.

**Reproducibility**. The paper includes the implementation details. The loss function change also should be quite straightforward. The reproducibility should be good.

**Strength And Weaknesses:**

### Strengths

1. The paper is quite well motivated at the high-level. It states clearly what issues to be addressed; how they propose to address them; and the experiments validate the proposal.

2. The related works are discussed quite thoroughly.

3. Experiments are conducted on three different tasks, which validate the effectiveness of the proposed method in more applications.

### Weaknesses

1. The paper is written based on the concept of "concordance". What is the formal definition of "concordance"? Can the conventional triplet loss in Eq(2) be considered a "concordance-induced triplet" loss? The paper presents several fancy loss functions, but the foundation does not seem to be that solid.

2. Some detailed tech choices are not well motivated or discussed. For example,

    1. Eq(4), why is it called the exponential lower-bound form? I assume it is because $1 - e^{-x} \leq x$?
    2. Eq(4), why do we have to take the exponential lower-bound form? Why cannot we just set the margin = 0 for the conventional triplet loss in Eq(2)?
    3. Eq(6), what is the "partial likelihood" for? Why cannot it be computed as $\frac{e^{S_{ap}}}{e^{S_{ap}} + e^{S_{an}} + e^{S_{pn}}}$ or other forms inspired by the cross-entropy softmax classification loss?
    4. The proposed CIT loss formula in Eq(8) reminds me of the classification + triplet + center loss used in [a]. How do they compare?

[a] Bag of Tricks and A Strong Baseline for Deep Person Re-identification, CVPRW 2019

3. There is a typo / inaccurate definition of the set of comparable pairs $E_T$ in section 2.2. Currently it enforces $S_{ap} > S{an}$, which does not seem to be right? Otherwise, the Eq(4) and Eq(5) will become meaningless.

4. The empirical significance of the proposed method is arguably marginal:

    1. In Table 1, the conventional triplet loss seems to be quite robust against the margin choices, and the results are all quite competitive. Then is tuning the margin really a trouble in practice? And how about we just set the margin = 0 as default?
    2. In Figure 3, $\gamma < 1.0$ enables the partial likelihood loss Eq (7) and $\gamma = 1.0$ disables it. However, the figures suggest that the convergence rates are not that different. The difference is slightly more obvious in ViT but still not very convincing.

**Summary Of The Paper:**

This paper aims at addressing two issues with the triplet loss: 1) needs to set a global violation margin, and 2) slow convergence during training. The paper proposes a "Concordance-Induced Triplet" (CIT) loss, which consists of two parts, one can be considered an exponential form of the conventional triplet loss with margin = 0; the other can be considered a variant of the softmax cross-entropy classification loss within each triplet.

Experiments on three tasks (person re-identification, image retrieval, face recognition) show that the proposed CIT loss leads to comparable accuracy and convergence rate with the conventional triplet loss, without having to tune the margin hyperparameter.

**Summary Of The Review:**

The paper presents some interesting improvements to the triplet loss, but as elaborated above, 1) it lacks some definitions/discussions to make the proposed improvements more theoretically grounded, and 2) the significance of the proposed method is arguably marginal according to the experiments. Thus, I would recommend a reject of the paper in its current version.

---

> ### Author Response · Authors · 2022-11-14
> **Response to Reviewer mdeV (Q1 and Q2)**
>
> $\textbf{Q1}$: (1) The paper is written based on the concept of "concordance". What is the formal definition of "concordance"? (2) Can the conventional triplet loss in Eq(2) be considered a "concordance-induced triplet" loss? (3) The paper presents several fancy loss functions, but the foundation does not seem to be that solid.
>
> $\textbf{A1}$:
>
> (1) With a comparable pair on a triplet sample, the intra-class similarity is naturally higher than the inter-class similarity. Suppose the predicted similarity also follows this relation. In that case, the predictive similarity relation of a comparable pair keeps consistency with the true similarity relation, thus being formulated as a concordance as the definition of the indicator function in Equation 5.
>
> (2) The triplet loss in Equation 2 exploits a decision margin to specify the similarity distance between a comparable pair, while our CIT loss only exploits concordance to constrain the similarity relation. If the margin m=0.0 in Equation 2, then the triplet loss can be considered a concordance-induced $L_{1}$ loss.
>
> (3) The concordance index defined in Equation 5 is the supremum of our CIT loss and it is a generalization of the Wilcoxon-Mann-Whitney statistics and thus of the area under the ROC curve (AUC) to regression problems [1,2]. With the concordance index, we present our CIT loss and its extension and discuss their generalization bound in Appendix A. The extensive experiments also confirm that our CIT can be an ideal optimizer for DML training when it is challenging to choose an appropriate margin out.
>
> $\textbf{Q2}$: Some detailed tech choices are not well motivated or discussed. (1) Eq(4), why is it called the exponential lower-bound form? (2) Eq(4), why do we have to take the exponential lower-bound form? Why cannot we just set the margin = 0 for the conventional triplet loss in Eq(2)? (3) Eq(6), what is the "partial likelihood" for? (4) The proposed CIT loss formula in Eq(8) reminds me of the classification + triplet + center loss used in \cite{luo2019bag}. How do they compare?
>
> $\textbf{A2}$:
>
> (1) The definition in Equation 5 is the concordance index, which can be lower-bounded by the exponential form in Equation 4.
>
> (2) Minimizing the concordance index of Equation 5 is a discrete optimization problem. Hence, we intend to devise a differentiable and convex lower bound on a 0-1 indicator function for gradient-based training. This function can be in an exponential form, log-sigmoid form, or other forms. In particular, by setting m=0.0, Triplet in Equation 2 can be viewed as a $L_{1}$ loss building on similarity concordance. Compared to Triplet of m=0.0, our CIT has a better generalization in an exponential form. There are two pieces of evidence: (a) Besides Figure 5, we also exhibit loss curves for Triplet on different margins in Figure 6. The loss value of Triplet of m=0.0 is infinitely close to zero quickly, thus being ineffective in driving hard triple samples to optimize DML models. (b) New experimental results reported in Tables 7 and 8 can demonstrate that our CIT stably outperforms Triplet of m=0.0.
>
> (3) The partial likelihood term builds on the triangle relation in our method and abides by the specification in [3].
>
> (4) Some training tricks in [4] are proposed to achieve high performances for ReID tasks, including an objective function combining ID, Triplet, and Center losses. Distinct from this objective function entirely, we exploit the concordant similarity relations of triplet samples to build a novel CIT loss for DML training, without tuning the decision margin. In our CIT loss, we apply the hyper-parameter $\gamma$ to regulate the degree of penalty on hard triplet samples.
>
> References:
>
> [1]Mann, Henry B., and Donald R. Whitney. "On a test of whether one of two random variables is stochastically larger than the other." The annals of mathematical statistics (1947): 50-60.
>
> [2]Wilcoxon, Frank. "Individual comparisons by ranking methods." Breakthroughs in statistics. Springer, New York, NY, 1992. 196-202.
>
> [3]Cox, David R. "Partial likelihood." Biometrika 62.2 (1975): 269-276.
>
> [4]Luo, Hao, et al. "Bag of tricks and a strong baseline for deep person re-identification." Proceedings of the IEEE/CVF conference on computer vision and pattern recognition workshops. 2019.

---

> ### Author Response · Authors · 2022-11-14
> **Response to Reviewer mdeV (Q3 and Q4)**
>
> $\textbf{Q3}$: There is a typo/inaccurate definition of the set of comparable pairs $\mathrm{E}_{\mathcal{T}}$ in section 2.2. Currently it enforces , which does not seem to be right? Otherwise, the Eq(4) and Eq(5) will become meaningless.
>
> $\textbf{A3}$: Thanks for your careful comments. We have revised the description of $\mathrm{E}_{\mathcal{T}}$ in section 2.2.
>
> $\textbf{Q4}$: The empirical significance of the proposed method is arguably marginal: (1) In Table 1, the conventional triplet loss seems to be quite robust against the margin choices, and the results are all quite competitive. Then is tuning the margin really a trouble in practice? And how about we just set the margin = 0 as default? (2) In Figure 3, $\gamma<1.0$ enables the partial likelihood loss Eq (7) and $\gamma=1.0$  disables it. However, the figures suggest that the convergence rates are not that different. The difference is slightly more obvious in ViT but still not very convincing.
>
> $\textbf{A4}$: In the long-term practice of DML training, we are plagued by task-specific hyper-parameters. The default value only is experienced-reference and is usually unable to apprise us where the supremum of performances is. Hence, we resort to tuning more margins or adopting more triplet-based loss functions. Undoubtedly, similarity concordance is the cornerstone of all triplet-based loss functions and motivates us to explore directly optimizing it. Our CIT loss favor DML training simply and elegantly by modeling the concordance of triplet similarity. Thus CIT can be a baseline of performance and free us from tuning hyper-parameters, contributing novelty and practicality for DML training.
>
> (1) Regarding the concern of tuning the margin, we provide more detailed reports on two different datasets in Tables 7 and 8, including performances of Triplet on m=0.0. Compared to Circle and Angular, Triplet and CT output stable performances on different values of the decision margin for the same task. But they obtain inconsistent performances on the same value for different tasks. In Table 7, Triplet achieves the best R@1 of 94.06 (m=0.05) and mINP of 61.94 (m=0.1) on ViT but not the best on CNN. In Table 8, CT obtains the best R@1 of 93.75 (m=0.05) and mAP of 79.17 (m=0.05) on CNN but not the best on ViT. In other words, the optimal value of the violate margin for a specific task is not competent for another task. Besides, in terms of mINP in Tables 7 and 8, Triplet and CT also emerge fluctuations on different margins for the same task.
>
> (2) The partial likelihood term $\mathcal{L}_{p}$ can impose more penalties on hard triplet samples, thus helping drive model optimization. Such efficacy varies on different backbones and different datasets with divergent sampling. As shown in Figures 1, 3, 5, and 6, we can observe that setting $\gamma<1.0$ can speed model convergence. Our CIT loss's significance is to provide an elegant and simple training optimizer for DML models. If the partial likelihood term can not significantly speed convergence, disabling this term also can not cause inferior performance.

---

### Official Review · Reviewer_FXz5 · 2022-10-25

**Confidence:** 4
**Correctness:** 4
**Technical Novelty And Significance:** 3
**Empirical Novelty And Significance:** 3
**Recommendation:** 5

**Clarity, Quality, Novelty And Reproducibility:**

The paper is well written, easy to follow and understanding. Codes are not available in the submission. The proposed method is novel to me, showing the property of elegance and simplicity. The overall quality of this is good, except for some weakness in experiments

**Strength And Weaknesses:**

**Strength**
1. The proposed CIT loss is new to me, and shows the advantages of elegance and simplicity.
2. The paper is easy to understand and well written. The motivation of this paper is solid.
3. Extensive experiments are conducted to demonstrate the superiority of the proposed method.

**Weakness**
1. In Eq. 4, does it need to tune for the term of ‘1’ in the range of (e$^{-1}$, e) or will there be gains from tuning it?
2. Figure 1 only shows the property of fast convergence compared with the variant of itself, how about compared with other losses (e.g., circle loss, soft triple loss) ?
3. Hard mining usually shows benefits to model training, but it seems inconsistent with the results from Tab.3, could the author provide some explanations or discussions?
4. Results in Tab. 1 and 2 are not always competitive compared with others. From Tab. 1, results of Circle and Angular are fluctuant with different values of margin, but those of Triplet and CT are much more stable. What are the values of margin used in Tab. 2. I am wondering, among these three tasks, if methods like Triplet or CT really need a careful tuning


**Summary Of The Paper:**

Conventional triplet-based losses require carefully tuning a decision boundary. To circumvent this issue, this paper proposes a novel yet efficient concordance-induced triplet (CIT) loss as an objective function to train deep metric learning model. Furthermore, it introduces a partial likelihood term for CIT loss to impose additional penalties on hard triplet samples, thus enforcing fast convergence. Extensive experiments are conducted on a variety of deep metric learning tasks to demonstrate the advantages of the proposed method.

**Summary Of The Review:**

This paper proposes a novel concordance-induced triplet (CIT) loss to train deep metric learning model, embracing the advantages of no need for margin tuning and fast convergence. However, there are some limitations as mentioned in the Weakness. I would like to hear the feedback from the authors.

---

> ### Author Response · Authors · 2022-11-11
> **Response to Reviewer FXz5**
>
> $\textbf{Q1}$: In Eq. 4, does it need to tune for the term of '1' in the range of $(e^{-1}, e)$, or will there be gains from tuning it?
>
> $\textbf{A1}$: With the term '1' in Equation 4, the loss range lies in $[1-e, 1-e^{-1}]$, which bounds on the $0-1$ indicator in Equation 5 by the hinge function. The term '1' is introduced to regulate loss range and not shift loss variance. Thus tuning it can not help advance gains.
>
> $\textbf{Q2}$: Figure 1 only shows the property of fast convergence compared with the variant of itself, how about compared with other losses (e.g., circle loss, soft triple loss) ?
>
> $\textbf{A2}$: The convergence speed is heavily related to the exploitation of hard triplet samples. But different losses vary in mining and exploiting hard triplet samples according to their specifications. Hence, we just compare the convergence speed between our CIT loss and its variant in Figure 1. In Figure 5, the left plot shows the convergence property of our CIT on $\gamma=1.0$ compared with other losses. It is observed that our CIT holds closer to Triplet and CT on convergence speed and stability.
>
> $\textbf{Q3}$: Hard mining usually shows benefits to model training, but it seems inconsistent with the results from Tab.3, could the author provide some explanations or discussions?
>
> $\textbf{A3}$: There are two terms in our CIT. The first term determines the final performance by mining hard triplet samples based on similarity concordance. By setting $\gamma<1.0$, we leverage the second term to enhance exploition on hard triplet samples to obtain fast convergence. Table 3 can prove that the second term is only used to raise convergence speed and can not help boost performance under complete sampling.
>
> $\textbf{Q4}$: (1) Results in Tables 1 and 2 are not always competitive compared with others. (2) From Table 1, results of Circle and Angular are fluctuate with different values of margin, but those of Triplet and CT are much more stable. What are the values of margin used in Table 2. I am wondering, among these three tasks, if methods like Triplet or CT really need a careful tuning.
>
> $\textbf{A4}$:
>
> (1) We acknowledge that our CIT is not always competitive with those losses with an appropriate violation margin. However, our CIT can be a qualified optimizer when the violation margin in conventional triplet losses is a burden during DML training. To be a qualified optimizer, our CIT should keep the first or closer to the first performance on various tasks. The experimental results on three datasets and two backbones are enough to prove that our CIT is a competent alternative.
>
> (2) The violate margin is task-specific, thus to be tuned carefully to get ideal performances. There are two essentials for tuning it carefully: (a) the same value presents inconsistent performances on different tasks, and (b) different values bring fluctuating performances on the same task. The reports about Circle and Angular in Table 1 can account for (b). Although CT and Triplet can gain relatively stable outputs on different values of margin compared to other losses, (a) occurs on them.
>
> In Table 1, Triplet achieves the best R@1 of 94.06 (m=0.05) and mINP of 61.94 (m=0.1) on ViT but not the best on CNN. In other words, the optimal value of the violate margin for a specific task is not competent for another task. To provide more convincing evidence, we extend the range of margins for Triplet and CT and report their performances in Table 7. The values of the violate margins for Triplet and CT are 0.05 in Table 2. In Table 8, we supplement their performance on different margins for the In-shop Clothes dataset. From the results in Table 8, we can observe that (a) occurs on CT. CT obtains the best R@1 of 93.75 (m=0.05) and mAP of 79.17 (m=0.05) on CNN but not the best on ViT. Besides, in terms of mINP in Tables 7 and 8, Triplet and CT also emerge (b).

---

### Official Review · Reviewer_TC7M · 2022-10-28

**Confidence:** 1
**Correctness:** 4
**Technical Novelty And Significance:** 2
**Empirical Novelty And Significance:** 3
**Recommendation:** 6

**Clarity, Quality, Novelty And Reproducibility:**

In general, the paper is well written and thus easy to read. As the tackled approach is feasible, it might be possible to re-implement the approach. There are just a few issues that need to be addressed to increase the readability and the clarity.

To increase the readability, remove the bullet points in the introduction.

Check the mathematical writing in Secs. 2.1 and 2.2.

Check the grammar and the terms in Sec.3.1.

The arrangement of tables and figures is hampering fluently reading the experimental section.

The plots in Fig.2 are too small to see the relevant details.

The plot-in-plot plots in Fig.3 are misleading.

The bibliography needs to be seriously checked for consistency, completeness, and correctness!

**Strength And Weaknesses:**

Strength:

The introduction gives a clear problem statements and a motivation for the proposed approach. In this way, also the related work is caputed in a meaningful way.

The technical description is simple but seems to be reasonable.

Weaknesses:

There is only a slight (if any) improvement compared to the used baselines?

**Summary Of The Paper:**

The paper tackles triplet loss learning by introducing a novel loss function: CIT. The approach is demonstrated for different applications.

**Summary Of The Review:**

Overall, the approach seems to be feasible and is based on theoretical foundation. The paper is well written and thus easy to read. On the downside, however, the experiments do not show clear benefits to the state-of-the-art.

---

> ### Author Response · Authors · 2022-11-09
> **Response to Reviewer TC7M**
>
> $\textbf{Q1}$: There is only a slight (if any) improvement compared to the used baselines?
>
> $\textbf{A1}$: Our CIT loss is a task-agnostic optimizer for deep metric learning. It can favor DML training simply and elegantly by modeling the concordance of triplet similarity. It is an optional DML training optimizer when existing triplet-based losses can not help DML models achieve ideal performances by tuning task-specific decision margins. Hence, we expect to make CIT achieve comparable performances with its counterparts and not deliberately pursue building new state-of-the-art.
>
> $\textbf{Q2}$: There are just a few issues that need to be addressed to increase the readability and the clarity.
>
> $\textbf{A2}$: Benefiting from your significant inputs and recommendations, we have been able to prepare a much better manuscript. Regarding the display issue of Figure 2, limited by space, we also provide relevant values in Tables 3, 4, and 5 (Appendix C).
>
> $\textbf{Q3}$: On the downside, however, the experiments do not show clear benefits to the state-of-the-art.
>
> $\textbf{A3}$: Experimental results on three tasks demonstrate that our CIT loss without the decision margin can achieve comparable performance with its state-of-the-art counterparts.

---

### Decision · Program_Chairs · 2023-01-20

**Decision:**

Reject

**Justification For Why Not Higher Score:**

The paper has significant weaknesses pointed out by the reviewers (even beyond marginal performance improvements).

**Justification For Why Not Lower Score:**

N/A

**Metareview: Summary, Strengths And Weaknesses:**

This paper proposes a Concordance-Induced Triplet (CIT) loss to train a metric learning-based model, aiming to avoid setting an absolute margin necessary for the prior methods with the motivation that its scale is dependent on factors such as intra-class variation. The loss specifically leverages the inter/intra-class similarity ordering. Experimental results are demonstrated across three retrieval-based datasets.

  While the reviewers thought the idea itself is interesting and solves a well-motivated problem, they all agreed that the paper is not ready for publication as-is. There were a number of issues identified including the writing, clarifications needed for the mathematical portion, figures, and bibliography. One of the key weaknesses identified across all of the reviewers is that the performance is only comparable, or marginally better in some cases, than the prior methods (and several methods such as the "bag of tricks" citation are not discussed/compared to). While state of art performance is not the only factor in deciding, there are several weaknesses that, combined with this, make the paper's contribution significant lessened. Specifically, the argument of hyper-parameter tuning is not convincingly shown given that there is a hyper-parameter introduced by the method, and to show that a single value can be used across many tasks it should be shown on a more comprehensive set of experiments, including potentially additional tasks beyond retrieval (as mentioned by reviewer nK2W.

  Overall, while the paper is promising, it unfortunately cannot be accepted as-is, as agreed by the reviewer consensus, and these weaknesses should be addressed for future submission.